# Extensive Genetic Diversity and Epidemiological Patterns of Factor H-Binding Protein Variants among *Neisseria meningitidis* in China

**DOI:** 10.3390/microorganisms12030481

**Published:** 2024-02-27

**Authors:** Zhizhou Tan, Juan Xu, Jie Che, Li Xu, Dongshan Yan, Maojun Zhang, Zhujun Shao

**Affiliations:** National Key Laboratory of Intelligent Tracking and Forecasting for Infectious Diseases, National Institute for Communicable Disease Control and Prevention, Chinese Center for Disease Control and Prevention, Beijing 102206, China; tanzhizhou@gmail.com (Z.T.); xujuan9270@163.com (J.X.); chejie@icdc.cn (J.C.); xuli@icdc.cn (L.X.); jesus0919@163.com (D.Y.); zhangmaojun@icdc.cn (M.Z.)

**Keywords:** *Neisseria meningitidis*, factor H-binding protein, vaccine

## Abstract

Factor H-binding protein (fHbp) is a virulence factor expressed by *Neisseria meningitidis* (*N. meningitidis*), the primary causative agent of invasive meningococcal disease (IMD) in humans. fHbp is utilized as the main component in vaccines to provide protection against IMD caused by serogroup B *N. meningitidis*. In order to comprehensively investigate the genetic diversity and epidemiological patterns of fHbp variants within isolates of Chinese *N. meningitidis*, we utilized the NEIS0349 locus, which encompasses the complete coding sequences of fHbp. This enabled us to identify allelic variants of fHbp with enhanced resolution. A total of 109 fHbp variants were identified in 1013 Chinese *N. meningitidis* isolates. We reconstructed a phylogenetic tree and analyzed the epidemiological characteristics of each variant. Considering both temporal and geographical distribution patterns, only four fHbp variants (v2.16, v2.18, v2.404, and v2.21) exhibited persistent nationwide prevalence during the previous decade (2011–2021). These variants were highly prevalent in both serogroup B strains from patients and healthy individuals, suggesting their potential as suitable vaccine candidates for nationwide implementation against IMD caused by serogroup B strains. Our study emphasizes the significance of conducting continuous surveillance of meningococcal strains to monitor the genetic diversity of fHbp for the purpose of vaccine development.

## 1. Introduction

*Neisseria meningitidis* (*N. meningitidis*) is the primary causative agent of invasive meningococcal disease (IMD) in humans [1,2]. Phenotypically, *N. meningitidis* is classified into 12 serogroups based on the polysaccharide structures of their capsules, among which six (A, B, C, W, X, and Y) are responsible for the majority of IMD cases worldwide [2]. Vaccines based on capsular polysaccharides as the main component can be used to control strains of serogroups A, C, W and Y for IMD [1,3]. However, due to its similarity to human neural cell adhesion molecule, the serogroup B capsular polysaccharide is poorly immunogenic and has the potential to trigger autoimmune disease [4]. Instead, the development of serogroup B meningococcal vaccines primarily focuses on subcapsular protein antigens.

*N. meningitidis* uses surface proteins to recruit human complement factor H and downregulates the alternative complement pathway, forming immune evasion mechanism to protect itself from complement-mediated killing [5,6,7]. Using two different approaches, proteins capable of recruiting H factor were identified as GNA1870/LP2086, later renamed as factor H-binding protein (fHbp) [8,9,10]. The discovery and importance of fHbp have prompted the development of two recombinant serogroup B vaccines, 4CMenB (Bexsero, GSK, London, UK) [11] and MenB-FHbp (Trumenba, bivalent rLP2086, Pfizer, New York, NY, USA) [12]. To date, both vaccines have been licensed and implemented in national immunization programs in many countries. Both vaccines are capable of eliciting good immunogenicity within the vaccinated population, inducing protective immune responses and being well tolerated [13,14,15,16].

The fHbp gene exhibits high genetic diversity in *N. meningitidis* [17,18]. The effectiveness of fHbp-based vaccines can be influenced by the selected fHbp variants included in the vaccines [10,19]. Of the two available vaccines, Bexsero contains fHbp variant 1.1 (v1.1), while Trumenba contains fHbp v1.55 and v3.45. However, these vaccines are not currently available in China, and it is unknown if they would be effective in the Chinese population. Therefore, further exploration is needed to determine if novel vaccines containing other fHbp variants should be developed [20]. It is crucial to investigate the extensive genetic diversity of fHbp allelic variants and their corresponding epidemiological patterns in Chinese *N. meningitidis* strains.

The pubMLST database provides two loci for the fHbp gene in *N. meningitidis*: one referred to as “fHbp” uses partial coding sequences (approximately 770 bp), while the other, NEIS0349, utilizes complete coding sequences (approximately 830 bp). Historically, allele variant names derived from the former locus have been widely used in fHbp studies [10,17,18,21]. However, the NEIS0349 locus, primarily used for core genome multilocus sequence typing (cgMLST) [22], presents several advantages over the “fHbp” loci: (1) The NEIS0349 typing allows for the determination of the completeness of the coding frame of the fHbp gene and the likelihood of encoding protein products. In contrast, the fHbp typing does not provide such information and could identify a pseudogene. (2) NEIS0349 typing, which utilizes complete coding sequences, demonstrates higher resolution compared to fHbp typing that employs partial coding sequences. (3) NEIS0349 typing requires the utilization of the complete genome of isolates, ensuring better retrospective analysis, reproducibility, and prevention of potential variant issues. For instance, fHbp variant v1.55, used in Trumenba vaccines [23], does not have a corresponding variant with a complete coding frame in *N. meningitidis* genomes found in the pubMLST database, suggesting the questionable nature of this variant.

In the present study, we employed NEIS0349 typing to identify the allelic variants of fHbp in a collection of 1013 Chinese *N. meningitidis* strains. This dataset represents the largest data of its kind in China to date. We conducted a comprehensive phylogenetic analysis, exploring the extensive genetic diversity of fHbp allelic variants and their corresponding epidemiological patterns in Chinese *N. meningitidis* strains.

## 2. Materials and Methods

### 2.1. Collecting of N. meningitidis Isolates and Multilocus Sequence Typing

A total of 1013 isolates were analyzed in this study. These isolated strains were obtained through meningococcal disease surveillance system of Chinese Centers for Disease Control and Prevention (CDC). This surveillance system collected isolates from invasive cases throughout China’s provinces, as well as isolates from healthy individuals [20]. Suspected *N. meningitidis* strains are collected from invasive cases by local CDCs. All collected strains are sent to the laboratory of the National Institute for Communicable Disease Control and Prevention (ICDC), Chinese CDC in Beijing for identification and further testing. Additionally, periodic surveys are also conducted to assess the healthy carriage of *N. meningitidis*, and carrier strains are isolated. These isolate strains have been stored at a temperature of −80 °C in skimmed milk. For each isolate, epidemiological information, such as the source of isolation, the year of collection, and the location, was included for analysis. MLST (multilocus sequence typing) of each isolate was performed using the FastMLST software v0.0.15 (https://github.com/EnzoAndree/FastMLST) accessed on 26 August 2022, with the pubMLST database, used as a reference. This allowed for the assignment of the sequence type (ST) and clonal complex (CC) to each isolate. The detailed information of these isolates is presented in Appendix A.

### 2.2. Serogrouping, DNA Preparation and Whole-Genome Sequencing

*N. meningitidis* strains were cultured on blood agar plates at 37 °C in a 5% CO_2_ atmosphere for 18–22 h. Gram staining and biochemical tests (API-NH, BioMerieux, Marcy L’Etoile, France) were used to confirm the isolates. Serogrouping was performed using slide agglutination (Remel, Lenexa, KS, USA). Pure cultures of *N. meningitidis* were eluted with saline solution, and the supernatant was discarded following centrifugation. Genomic DNA extraction was carried out using the Genomic DNA Purification Kit (Promega, Madison, WI, USA) as instructed by the manufacturer. The purity and integrity of the DNA were assessed and quantified using the Agilent 5400 system. Genomic DNA that met quality control standards was subjected to whole-genome sequencing on the Illumina NovaSeq PE150 platform to obtain the draft genome. The sequencing was conducted by Beijing Novogene Corporation Ltd in Beijing city, China.

### 2.3. Genome Validation and Quality Control

All the genomes of isolates were validated by the species identification function (taxonomy_wf) in Genome Database Taxonomy toolkit (GTDB-Tk) [24]. All isolates were compared with a representative *N. meningitidis* genome, and the average nucleotide identity (ANI) values were calculated for each isolate. A threshold of 0.95 ANI was used to confirm that the genomes belonged to the *N. meningitidis* species. The quality of the genomes was assessed using CheckM software v1.2.1 [25]. This evaluation involved evaluating the completeness and contamination of each genome. Of the isolates, only those with a completeness rate of ≥90% and a contamination rate of ≤5% were considered suitable for further analysis.

### 2.4. Detection and Genotyping of fHbp Allelic Variants

The pubMLST database provides two loci for the fHbp gene: one locus, called “fHbp”, utilizes partial coding sequences of fHbp, while another locus, NEIS0349, utilizes complete coding sequences of fHbp. However, the allele variant names in the fHbp locus did not correspond to the NEIS0349 locus. This limitation hinders the use of NEIS0349 typing assay. We utilized the vmatchPattern function in the R package Biostrings v2.68.1 (https://github.com/Bioconductor/Biostrings) accessed on 16 November 2023 to match each fHbp variant to its respective NEIS0349 variant(s).

The FASTA files for the fHbp locus and NEIS0349 locus were retrieved from the pubMLST database as of Oct 31th, 2023. To establish a schema, the FASTA file of the NEIS0349 locus was processed using the PrepExternalSchema module in chewBBACA software v2.8.5 [26]. Subsequently, the NEIS0349 variants in 1013 Chinese *N. meningitidis* isolates were identified using the AlleleCall module in chewBBACA software v2.8.5.

### 2.5. Association Analysis of fHbp Variants, Serogroups, and Clonal Complexes

In order to measure the association between fHbp variants, serogroups, and clonal complexes, Cramer’s V was calculated by the assocstats function in the R package vcd 1.4-12 [27].

### 2.6. Phylogenetic Analysis and Identifying Epidemiological Patterns of fHbp Allelic Variants

The maximum likelihood tree of NEIS0349 variants was reconstructed based on the aligned nucleotide sequences of variants using the IQ-TREE v2.2.0.3 [28]. The best-fit substitution model TIM + F+I + G4 identified by jModelTest was implemented in IQ-TREE. The branch supports were assessed by ultrafast bootstrap approximation with 1000 replicates. The tree and epidemiological characteristics of NEIS0349 variants were visualized and annotated by R package ggtree v3.6.2 [29] and ggtreeExtra v1.6.1 [30].

## 3. Results

### 3.1. fHbp Allelic Variants Existed in Most Chinese N. meningitidis Strains with Extensive Genetic Diversity

A total of 1013 *N. meningitidis* isolates, collected from 1956 to 2021 across China, were included in the analysis (Appendix A). Among them, 35 isolates were collected after 2019, representing the Chinese strains that spread following the onset of the coronavirus disease 2019 (COVID-19) pandemic. This collection of isolates represents the largest dataset of China to date, exhibiting a higher phenotypic diversity (8 serogroups, NG serogroup and *cnl* strain) and significant genetic diversity (18 clonal complexes and numerous UA strains). All isolates were sequenced using the NGS method, and draft genomes were obtained. If present, the CDS of fHbp allelic variants (NEIS0349) was extracted from each genome.

The fHbp allelic variants were present in a majority of Chinese *N. meningitidis* strains, while exhibiting a high level of genetic diversity. The CDS of fHbp sequences has been successfully identified in 988 Chinese *N. meningitidis* isolates, only 25 isolates were found to be lacking the fHbp gene. As shown in Figure 1D and Figure 2, the fHbp gene displays significant diversity, with a total of 109 allelic variants (NEIS0349). The majority of allelic variants have low frequencies, with 76 variants occurring in fewer than 3 isolates. In contrast, the remaining 33 variants have higher frequencies, ranging from 215 to 4 isolates. A total of 56 new allelic variants in NEIS0349 loci, marked as NEIS0349_1644 to NEIS0349_1710) in Figure 1D, were also discovered and all of these novel variants are of low frequency.

The NEIS0349 typing can correspond to fHbp typing. Due to NEIS0349 typing utilizing complete coding sequences while fHbp typing utilizes partial coding sequences, NEIS0349 typing demonstrated higher resolution than fHbp typing, and each fHbp variant could correspond to multiple NEIS0349 variants. In Figure 1A, it can be observed that the most predominant variant of fHbp was v2.16, accounting for 21.2% of the isolates. Among the top ten fHbp variants, consisting of two variants in family 1, seven variants in family 2, and one variant in family 3; they were identified in more than 67% of the isolates, ranging from 217 to 29 isolates, respectively.

Generally, NEIS0349 variants that correspond to the same fHbp variant are genetically closely related. For instance, a monophyletic group (Figure 2) was formed by seven NEIS0349 variants corresponding to fHbp v3.45. However, one fHbp variant can have multiple NEIS0349 variants with lower genetic similarity and distinct epidemiological characteristics. The clearest example of this is fHbp v2.404, which corresponds to two NEIS0349 variants (NEIS0349_15 and NEIS0349_641). These two variants occupy different lineages in the phylogenetic tree (Figure 2). The NEIS0349_641 variant is mostly found in strains of CC4821, whereas the NEIS0349_15 variant is exclusively identified in ungrouped strains with phenotypic serogroup B (Figure 1C,D).

The pathogenicity of different fHbp variants may vary, and certain variants are predominantly found in strains isolated from patients with meningitis. Among these variants, six variants are primarily isolated from patients, accounting for over 60% of the cases. These variants are v1.5 (NEIS0349_81), v1.38 (NEIS0349_165), v1.419 (NEIS0349_642), v1.80 (NEIS0349_1664), v2.16 (NEIS0349_59), and v3.1239 (NEIS0349_1652). It is worth noting that among the NEIS0349 variants, v2.16 (NEIS0349_59) was predominantly identified in patients, whereas v2.16 (NEIS0349_60) exhibited a significantly lower incidence (Figure 1B).

### 3.2. The Phylogenetic Tree Displayed the Phenotypic and Genetic Characteristics of fHbp Allelic Variants in Chinese N. meningitidis Strains

As exhibited in Figure 2, the maximum likelihood method was used to reconstruct the phylogenetic tree of Chinese strains’ fHbp alleles. The topological structure of the tree was verified 1000 times, and nodes with bootstrap values greater than 70% and 50% (marked in red and pink, respectively) are considered highly reliable. It is evident that the fHbp alleles of Chinese strains form two major evolutionary branches (named as clade 1 and clade 2, respectively). Clade 1 corresponds to fHbp variant family 1, while clade 2 corresponds to fHbp variant family 2 and 3 [9]. The reconstructed tree includes not only all the fHbp allele variants in Chinese strains but also the fHbp allele variants found in Bexsero (v1.1) and Trumenba (v3.45) vaccines, which are two protein-based serogroup B meningococcal vaccines that are currently available [11]. It is evident that the v1.1 and v3.45 variants are virtually absent in Chinese strains.

The frequency analysis of these alleles in different serogroups and clonal complexes, attached to the phylogenetic tree, revealed the phenotypic and genetic characteristics of fHbp allelic variants in Chinese *N. meningitidis* strains (Figure 2). Phenotypically, the serogroup B, NG, and C strains exhibited the highest genetic diversity of fHbp, with 86, 33, and 22 variants, respectively. In contrast, other serogroups (including *cnl* strains) generally demonstrated limited genetic fHbp diversity, ranging from 1 to 9 variants. Genetically, the UA and CC4821 strains displayed the highest genetic diversity of fHbp, with 67 and 34 variants, respectively. On the other hand, other clonal complexes generally exhibited limited genetic fHbp diversity, ranging from 1 to 8 variants.

It is clearly shown that the fHbp variants from the same serogroups or clonal complexes do not cluster together in the tree. The fHbp variants of serogroup B and CC4821 are scattered in clade 1 and clade 2 and can be classified as fHbp variant families 1, 2, and 3 (in Figure 2). This observation suggests the presence of extensive homologous recombination events in the evolutionary history of fHbp. However, connections can still be observed between certain fHbp variants and serogroups or clonal complexes. For example, in Figure 1C,D, v1.5 (NEIS0349_81) and v1.38 (NEIS0349_165) are predominantly present in CC5 and CC1, which phenotypically belong to serogroup A, while v2.151 (NEIS0349_130) is predominantly present in CC11, which phenotypically belongs to serogroup W. The association between fHbp variants, serogroups, and clonal complexes was assessed using Cramer’s V statistic, a measure based on chi-square analysis. Cramer’s V value for the association between fHbp variants and serogroups was 0.574, indicating a moderate association. On the other hand, Cramer’s V value for the association between fHbp variants and clonal complexes was 0.781, indicating a strong association.

Due to the occurrence of extensive homologous recombination events in the evolutionary history of fHbp, further identification was conducted to determine the phenotypic and genetic characteristics of specific fHbp variants. Specifically, the top ten most prevalent fHbp variants were subjected to detailed investigation (Figure 2). The most prevalent fHbp variants in Chinese strains were v2.16 (NEIS0349_60) and v2.18 (NEIS0349_67) of the fHbp variant family 2. These two variants were also the most prevalent fHbp variants in serogroup B strains, representing 29.0% and 11.8% of the strains, respectively. However, they exhibited clear difference of presence in other serogroups and clonal complexes. v2.16 (NEIS0349_60) was the most prevalent variant in serogroup NG strain (17.0%) and also the most prevalent variant in CC4821 (54.9%), while v2.18 (NEIS0349_67) was the most prevalent variant in serogroup Y strain (61.2%) and also the most prevalent variant in CC175 (36.0%).

The lesser prevalent fHbp variants in Chinese strains were v2.404 (NEIS0349_15), v2.21 (NEIS0349_7), v2.22 (NEIS0349_1), v2.19 (NEIS0349_65), v2.101 (NEIS0349_13), and v2.404 (NEIS0349_641) of the fHbp variant family 2. These variants were present in serogroup B strains at frequencies ranging from 6.0% to 2.5%, respectively. While these variants showed similar prevalence in serogroup B strains, they exhibited clear differences in their presence within clonal complexes. Specifically, v2.101 (NEIS0349_13) was the most prevalent variant in CC32 (86.2%), whereas v2.19 (NEIS0349_65) was the most prevalent variant in CC41/44 (67.8%).

There were also three highly prevalent variants that mainly present in serogroup A or *cnl* strains. The v1.5 (NEIS0349_81) and v1.38 (NEIS0349_165) in fHbp variant family 1 were the most prevalent variants in serogroup A (60.0% and 30.0%, respectively). While v3.94 (NEIS0349_1645) in fHbp variant family 3 were the most prevalent variants in *cnl* strains (100%).

### 3.3. The Geographical and Temporal Distribution Patterns of fHbp Allelic Variants in Chinese N. meningitidis Strains Uncovered Potential fHbp Types for Vaccine Development

Additional analyses were conducted to assess the geographical and temporal distribution patterns of fHbp allelic variants among Chinese *N. meningitidis* strains. The primary objective was to evaluate the appropriateness of the most prevalent fHbp variants as potential vaccine candidates for national-scale and up-to-date immunization efforts.

As shown in Figure 3, the isolates included in the present study were collected from 29 provinces across China. Due to the vast and diverse landscape of China, these provinces were categorized into six geographical regions: North China, East China, South Central China, Southwest China, Northwest China, and Northeast China. Frequency analysis of fHbp variants revealed that fHbp variant family 3 was almost nonexistent in Southwest and Northwest China, whereas variants from fHbp variant families 1 and 2 were distributed across all six regions.

The frequency analysis also demonstrated that only eight fHbp allelic variants were reported across all regions, with six variants belonging to fHbp variant family 2 and two variants belonging to family 1 (Figure 3). Based on the analysis presented in the preceding section, it was observed that the six fHbp variants belonging to family 2 (v2.16, v2.18, v2.21, v2.23, v2.404, and v2.22) were the prevalent variants in serogroup B strains, whereas the two fHbp variants from family 1 (v1.5 and v1.38) were the prevalent variants in serogroup A strains.

Temporal patterns of fHbp allelic variants were identified by analyzing the time range and occurrence frequency of each variant in each year. As illustrated in Figure 4, several fHbp allelic variants dispersed throughout the phylogenetic tree have demonstrated persistence for over ten years, with v2.21 (NEIS0349_7) lasting for 56 years, and v1.5 (NEIS0349_81) lasting for 55 years. However, the majority of variants are transient and do not continue to spread. For instance, out of the 31 variants that were initially identified after 2011, only 1 variant, v2.572 (NEIS0349_1648), exhibited persistence for more than 2 years.

Effective vaccines should use currently prevalent variants rather than historically abundant but now missing variants. We specifically studied variants that were prevalent and widespread during the past decade (2011–2021), with particular emphasis on those that emerged following the onset of the COVID-19 pandemic (2019–2021). The analysis of time range and occurrence frequency represents eight fHbp allelic variants that have persisted for over ten years and continue to exist after 2019 (Figure 4). Among these variants, five belong to fHbp variant family 2, two belong to family 1, and one belongs to family 3. Based on the analysis presented in the preceding section, it was observed that the five fHbp variants belonging to family 2 (v2.16, v2.18, v2.21, v2.101, and v2.404) were the predominant variants in serogroup B strains. In contrast, the two fHbp variants from family 1 (v1.80 and v1.571) and one variant from family 3 (v3.1239) were not highly prevalent in any serogroups, thereby suggesting that these variants may exhibit sustained transmission at a lower level.

The v1.571 (NEIS0349_1121) were identified in serogroup B and NG strains that cannot assign to any clonal complex. However, this variant is not a dominant variant among the B strains, accounting for only 1.4%, and it has only been found in the healthy population so far. In contrast, although v3.1239 (NEIS0349_1652) and v1.80 (NEIS0349_1664) also have low frequencies of occurrence and limited geographical distribution, they are almost exclusively found in patients with meningitis. Among them, v3.1239 was only found in patients infected with serogroup B strains in Guangdong, Hunan, and Hebei provinces. On the other hand, v1.80 is distributed in the eastern coastal and northeastern regions and is primarily found in patients infected with serogroup B and C strains.

Apart from the eight clearly identified variants that have been present in the last decade, there is also a variant, v2.19, which is quite unique and deserves special attention. The v2.19 can be typed into two NEIS0349 variants: one variant v2.19 (NEIS0349_65) was initially identified in 1977 but has not been identified since 2012, whereas another variant v2.19 (NEIS0349_83) was only discovered in 2021. The significant temporal differences in the distribution of these variants belonging to the same genotype further emphasize the necessity of using CDS for typing fHbp.

## 4. Discussion

Our collection of isolates represents the largest dataset in China to date for identifying fHbp allelic variants in *N. meningitidis*. We utilized complete coding sequences of fHbp for analysis, which revealed detailed epidemiological characteristics of each variant. This provides valuable data for the development of potential fHbp-based vaccine candidates against meningococcal disease caused by *N. meningitidis*.

Considering both the temporal and geographical distribution patterns, only four long-lasting fHbp variants (v2.16, v2.18, v2.404, and v2.21) spread nationwide and remained in circulation beyond 2019. These variants exhibited a high prevalence in serogroup B strains and were found in both patients and healthy individuals, indicating their potential suitability as vaccine candidates for nationwide implementation against meningococcal disease caused by serogroup B strains. Among these four variants, v2.16 predominantly targets CC4821 strains and offers coverage against NG serogroup strains. On the other hand, v2.18 demonstrates a preference for CC175 and provides coverage for the Y serogroup. The v2.404 can be divided into two subtypes based on CDS, with one subtype NEIS0349_15 primarily present in serogroup B strains from UA strains and the other subtype NEIS0349_641 mainly found in CC4821 strains, providing coverage for both the B and W serogroups strains. v2.21 is mainly present in UA strains.

The v2.101 variant is also predominantly found in serogroup B strains and has been continuously prevalent in the past decade. However, it has not been discovered to be distributed in the northwest China. This specific variant mainly targets CC32 and requires strengthened monitoring in the northwest China to confirm if the current data reflect the actual geographical distribution.

Similarly, the v2.22, v2.23, and v2.19 variants were predominantly found in serogroup B strains. However, the spread duration of these variants does not meet the requirements of an ideal vaccine candidate. The v2.22 and v2.23 variants are distributed nationwide, but they have not been detected since 2018 and 2007, respectively. The v2.19 variant is not present in northeast China and can be classified into two subtypes based on CDS. Subtype NEIS0349_65 has not been identified since 2012, whereas subtype NEIS0349_83 was only discovered in 2021. Continual monitoring of these variants is crucial to confirm whether they have truly disappeared.

The v3.1239 and v1.80 are both variants that predominantly exist in CC4821 strains and are primarily found in patients with meningitis. Although they occur with low frequency and have limited geographical distribution, they have been consistently spread in the past decade. Among them, v3.1239 has only been detected in the three provinces of Guangdong, Hunan, and Hebei, but in all cases, it was found in patients infected with serogroup B strains. On the other hand, v1.80 is distributed in the eastern coastal and northeastern regions of China and primarily found in patients infected with both serogroup B and C strains. These two variants can potentially serve as vaccine candidates and be disseminated among susceptible populations to reduce the incidence rate of meningitis.

IMD poses a significant global public health concern. In China, from 2015 to 2019, the incidence rate of IMD was 0.078 per million individuals, with a case fatality rate of 11.82% [31]. The majority of reported cases in China affect individuals aged 10–19 years, followed by those aged 1–9 years [32]. Serogroups B, C, and W have been identified as the main causes of IMD in China from 2010 to 2020. After 2015, the prevalence of serogroup B increased to 52.4% [33]. Regrettably, China’s routine immunization program solely offers the meningococcal A polysaccharide vaccine and the combined meningococcal A/meningococcal C polysaccharide vaccine [34]. The present distribution of serogroups underscores the critical need to integrate meningococcal B vaccines into China’s National Immunization Program.

As polysaccharide-based vaccines for meningococcal B are unavailable, protein-based vaccines are considered a reliable option. The licensed meningococcal B vaccines, Bexsero and Trumenba, fall under this category, with fHbp serving as a crucial antigen in both formulations [35,36]. Given the significant discrepancy between the fHbp variants in China and those utilized in these vaccines, along with recent data from the Meningococcal Antigen Typing System (MATS) indicating that 4CMenB offers limited coverage for the fHbp v2.16 strains [37], the integration of variant-specific fHbp (v2.16, v2.18, v2.404, and v2.21) into new or existing meningococcal B vaccine formulations is imperative in China.

The main limitation of our study is the inherent sampling bias. The *N. meningitidis* isolates examined in the present study were obtained from the IMD surveillance network established by the Chinese CDC and provincial CDCs [38]. The primary flaw of this network is its dependence on passive surveillance, lacking active monitoring of *N. meningitidis* carriages, which can be as high as 15.50% in some regions of China [31]. Although we incorporated a substantial number of isolates from carriages in this study (801 out of 1013), our sampling still inevitably skews towards data from patients, leading to sampling bias. Furthermore, our sampling also exhibits temporal and spatial biases, with a predominant collection of strains post-2000 and limited samples from western China. These sampling biases may underestimate the diversity of fHbp and overlook some persistent variants. Future studies should enhance nationwide surveillance through regular monitoring of healthy populations to mitigate sampling biases.

Our research outlines specific future research directions. First, it is imperative to standardize laboratory operations across all levels, enhance monitoring, isolate bacterial strains, and conduct whole-genome sequencing. In monitoring antigen genes like fHbp, it is essential to utilize whole-genome data for improved reproducibility and traceability, with gene diversity analysis relying on full coding sequences of antigen genes rather than partial ones. Second, continuous, long-term, and cross-regional data are crucial for monitoring highly variable antigen genes like fHbp. While new variants emerge, the majority are transient; thus, a foundation for vaccine development and immunization planning can only be established through sustained monitoring. Third, the development of meningococcal B vaccines in China should prioritize the major domestic fHbp variants of serogroup B strains.

## Figures and Tables

**Figure 1 microorganisms-12-00481-f001:**
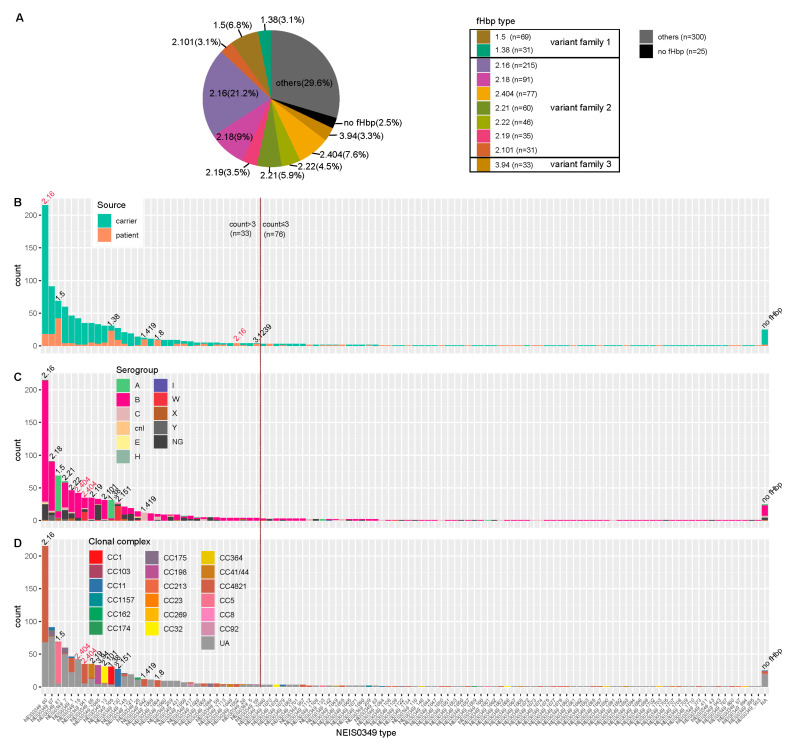
fHbp typing (**A**) and NEIS0349 typing (**B**–**D**) of 1013 Chinese *N. meningitidis* isolates. (**A**) This pie chart illustrates the top 10 fHbp variants based on isolate counts, which can be classified into three fHbp variant families. (**B**–**D**) These histograms illustrate NEIS0349 variants (n = 109) identified in Chinese isolates based on isolate counts. Among all of them, a total of 33 variants were identified in more than 3 isolates. These bar plots are colored by isolation source (**B**), serogroups (**C**) and clonal complexes (**D**). Each NEIS0349 variant was annotated with the corresponding fHbp variant on the respective bar, indicating the presence of similar characteristics among the majority of isolates in that bar (more than 60% isolates). The fHbp variants colored in red indicate that these particular variants can correspond to multiple NEIS0349 variants, each exhibiting distinct characteristics.

**Figure 2 microorganisms-12-00481-f002:**
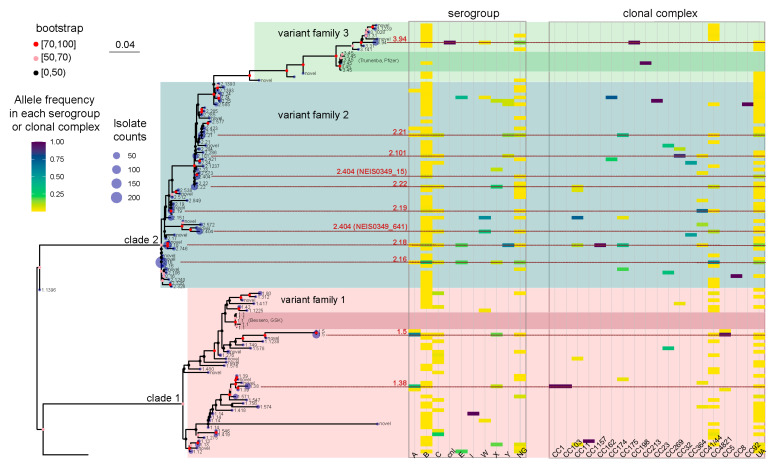
A maximum likelihood tree was constructed to represent the fHbp allelic variants using the 109 NEIS0349 variants. The number of isolates for each variant identified in the 1013 Chinese *N. meningitidis* isolates was displayed at the tips of the tree. The branch supports were evaluated using ultrafast bootstrap approximation with 1000 replicates. Two heatmaps located in the right panel of the tree was used to display the phenotypical and genetic information of each NEIS0349 variant, including their serogroups and clonal complexes, with the heatmap corresponding to the frequency of occurrence of variants within each serogroup or clonal complex. The top 10 NEIS0349 variants, based on isolate counts, were highlighted by red lines and colored in red, along with their corresponding fHbp variant.

**Figure 3 microorganisms-12-00481-f003:**
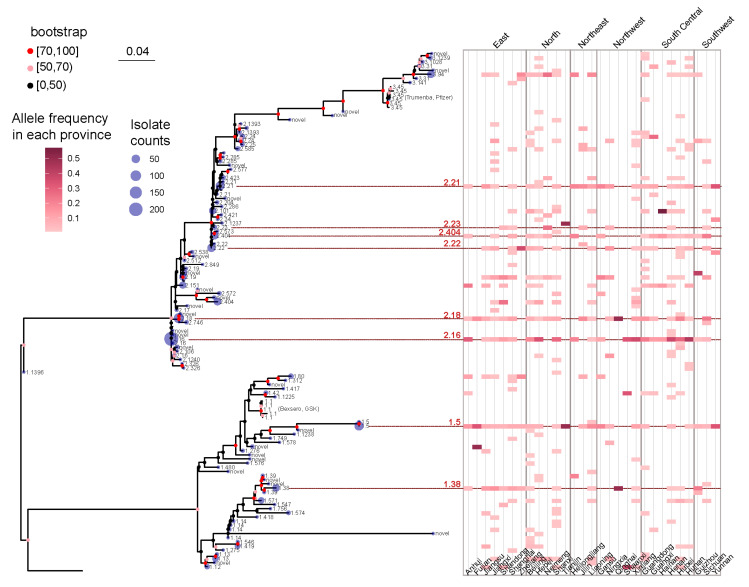
A maximum likelihood tree was constructed to represent the fHbp allelic variants using the 109 NEIS0349 variants. The number of isolates for each variant identified in the 1013 Chinese *N. meningitidis* isolates was displayed at the tips of the tree. The branch supports were evaluated using ultrafast bootstrap approximation with 1000 replicates. A heatmap, positioned in the right panel of the tree, was used to display the geographical distribution information of each NEIS0349 variant, including their collected location at the provincial level. The heatmap corresponds to the frequency of occurrence of variants within each province of China. The provinces were categorized into six geographical regions: North China, East China, South Central China, Southwest China, Northwest China, and Northeast China. Eight fHbp allelic variants were reported across all regions were highlighted by red lines and colored in red, along with their corresponding fHbp variant.

**Figure 4 microorganisms-12-00481-f004:**
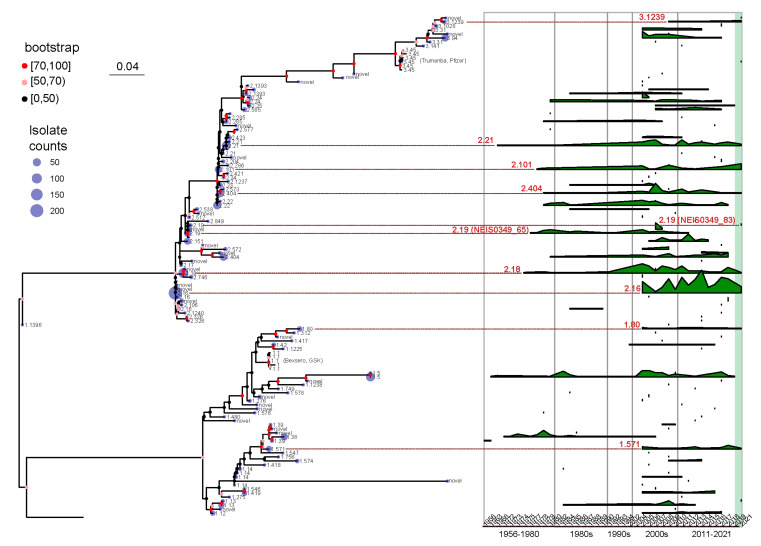
A maximum likelihood tree was constructed to represent the fHbp allelic variants using the 109 NEIS0349 variants. The number of isolates for each variant identified in the 1013 Chinese *N. meningitidis* isolates was displayed at the tips of the tree. The branch supports were evaluated using ultrafast bootstrap approximation with 1000 replicates. A ridgeline plot, positioned in the right panel of the tree, was utilized to visualize the temporal distribution information of each NEIS0349 variant, based on the collected years of isolates. The plot represents the isolate counts of each variant in each year. Additionally, nine fHbp allelic variants reported across all regions were highlighted by red lines and colored in red, along with their corresponding fHbp variant.

## Data Availability

All data available in the manuscript and the Appendix A.

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
