# Peer review of "Extensive Genetic Diversity and Epidemiological Patterns of Factor H-Binding Protein Variants among Neisseria meningitidis in China"

_microorganisms, 2024, doi:10.3390/microorganisms12030481_

Round 1

Reviewer 1 Report

Comments and Suggestions for Authors

A very interesting manuscript investigating the geographical distribution patterns of the Factor H Binding Protein (fHBP) in N. meningitidis. The authors found four fHbp variants (v2.16, v2.18, v2.404, and v2.21) that exhibited nationwide persistence. The authors proposed the use of these variants present in serogroup B strains from patients and healthy individuals, as suitable vaccine candidates.

The results were clearly presented and the quality of the graphs could be improved but that is the nature of these graphs. 

Author Response

Comments 1: A very interesting manuscript investigating the geographical distribution patterns of the Factor H Binding Protein (fHBP) in N. meningitidis. The authors found four fHbp variants (v2.16, v2.18, v2.404, and v2.21) that exhibited nationwide persistence. The authors proposed the use of these variants present in serogroup B strains from patients and healthy individuals, as suitable vaccine candidates. The results were clearly presented and the quality of the graphs could be improved but that is the nature of these graphs.

Response: Thank you for your review and suggestions. Due to the submission system's recommendation to use a template, the clarity of images in the manuscript is somewhat lacking. To address this deficiency, we have also uploaded PDF files of each figure to the submission system. These PDF files are high quality, and we hope the image editors can replace the low-resolution figures within the manuscript in the published version.

Reviewer 2 Report

Comments and Suggestions for Authors

The manuscript details an in-depth study on the genetic diversity of Factor H Binding Protein (fHbp) in Neisseria meningitidis strains across China. It highlights the identification of 109 fHbp variants from 1013 isolates, emphasizing a relatively small subset that exhibited consistent prevalence nationwide over ten years. The study emphasizes the significance of these findings for vaccine development, particularly against serogroup B meningococcal infections, advocating for ongoing surveillance to track fHbp variant distribution and evolution. This work is important for informing future vaccine strategies and understanding the epidemiological dynamics of N. meningitidis in China. This manuscript is well written. The topic is slightly outside of my expertise. Below are some general comments.

1.      It would be beneficial to include a section discussing the implications of the identified fHbp variants on the current and future landscape of meningococcal B vaccine development. This could involve a comparison with antigens targeted by existing vaccines, potential cross-protection capabilities, and considerations for incorporating variant-specific antigens into new or existing vaccine formulations.

2.      The manuscript could benefit from a broader discussion on the epidemiology of Neisseria meningitidis in China, including trends in disease incidence and the impact of vaccination programs on disease prevalence.

3.      Clearly articulating the study's limitations, such as the geographic and temporal scope of the isolate collection, potential biases in the selection process, and the limitations inherent in the methods used for variant identification, would strengthen the manuscript. Discussing these limitations and their potential impact on the study's conclusions will help readers critically assess the robustness of the findings.

4.      The manuscript could outline specific future research directions, such as the need for longitudinal studies to monitor the evolution of fHbp variants over time, the development of novel vaccines targeting the most prevalent and persistent variants, and the implementation of genomic surveillance programs to detect emergent variants. Suggesting how these findings could guide public health strategies and vaccine policy would provide valuable insights for future efforts in meningococcal disease prevention.

Comments on the Quality of English Language

A thorough proofreading could enhance the manuscript by identifying and correcting any minor grammatical errors, typographical mistakes, or inconsistencies in terminology usage. Attention to detail in these areas will further improve the overall quality and readability of the document.

For instance, "Factor H Binding Protein" is sometimes referred to as "fHbp" and other times in its full form. Decide on one format and stick to it for the entire document to maintain consistency.

Review the manuscript for proper use of hyphenation in compound adjectives. For example, "high genetic diversity" does not require a hyphen, but compound modifiers like "long-lasting fHbp variants" do. Ensure these rules are uniformly applied.

Check for proper use of commas in compound sentences and ensure that clauses are correctly separated. For example, "In order to comprehensively investigate the genetic diversity and epidemiological patterns of fHbp variants within isolates of Chinese N. meningitidis, we utilized the NEIS0349 locus..." The comma after "meningitidis" correctly separates the introductory phrase from the main clause.

Upon the first mention of an acronym, the term should be spelled out with the acronym in parentheses. Subsequent mentions can use the acronym alone. Verify that this rule is consistently applied, especially for terms like NEIS0349, MLST (Multilocus Sequence Typing), and others relevant to your study.

Author Response

Comments 1: It would be beneficial to include a section discussing the implications of the identified fHbp variants on the current and future landscape of meningococcal B vaccine development. This could involve a comparison with antigens targeted by existing vaccines, potential cross-protection capabilities, and considerations for incorporating variant-specific antigens into new or existing vaccine formulations.

Response: Thanks for your suggestions about the discussion of the manuscript. We have added a section discussing the meningococcal B vaccines and fHbp variants, see lines 400 to 408:

“As polysaccharide-based vaccines for meningococcal B are unavailable, protein-based vaccines are considered a reliable option. The licensed meningococcal B vaccines, Bexsero and Trumenba, fall under this category, with fHbp serving as a crucial antigen in both formulations. Given the significant discrepancy between the fHbp variants in China and those utilized in these vaccines, along with recent data from the Meningococcal Antigen Typing System (MATS) indicating that 4CMenB offers limited coverage for the fHbp v2.16 strains, the integration of variant-specific fHbp (v2.16, v2.18, v2.404, and v2.21) into new or existing meningococcal B vaccine formulations is imperative in China.”

Comments 2: The manuscript could benefit from a broader discussion on the epidemiology of Neisseria meningitidis in China, including trends in disease incidence and the impact of vaccination programs on disease prevalence.

Response: We wholeheartedly align with your viewpoints. A section has been incorporated to address the epidemiological aspects of Neisseria meningitidis in China and China's National Immunization Program concerning meningococcal disease, see lines 390 to 399:

“IMD poses a significant global public health concern. In China, from 2015 to 2019, the incidence rate of IMD was 0.078 per million individuals, with a case fatality rate of 11.82%. The majority of reported cases in China affect individuals aged 10-19 years, followed by those aged 1-9 years. Serogroups B, C, and W have been identified as the main causes of IMD in China from 2010 to 2020. After 2015, the prevalence of serogroup B increased to 52.4%. Regrettably, China's routine immunization program solely offers the meningococcal A polysaccharide vaccine and the combined meningococcal A/meningococcal C polysaccharide vaccine. The present distribution of serogroups underscores the critical need to integrate meningococcal B vaccines into China's National Immunization Program.”

Comments 3: Clearly articulating the study's limitations, such as the geographic and temporal scope of the isolate collection, potential biases in the selection process, and the limitations inherent in the methods used for variant identification, would strengthen the manuscript. Discussing these limitations and their potential impact on the study's conclusions will help readers critically assess the robustness of the findings.

Response: The primary constraint of our research lies in the intrinsic sampling bias, which might lead to an underestimation of the diversity of fHbp and the oversight of certain persistent variants. A section has been incorporated in the discussion to address the study's limitations, see lines 409 to 420:

“The main limitation of our study is the inherent sampling bias. The N. meningitidis isolates examined in the present study were obtained from the IMD surveillance network established by the Chinese CDC and provincial CDCs. The primary flaw of this network is its dependence on passive surveillance, lacking active monitoring of N. meningitidis carriages, which can be as high as 15.50% in some regions of China. Although we incorporated a substantial number of isolates from carriages in this study (801 out of 1013), our sampling still inevitably skews towards data from patients, leading to sampling bias. Furthermore, our sampling also exhibits temporal and spatial biases, with a predominant collection of strains post-2000 and limited samples from western China. These sampling biases may underestimate the diversity of fHbp and overlook some persistent variants. Future studies should enhance nationwide surveillance through regular monitoring of healthy populations to mitigate sampling biases.”

Comments 4: The manuscript could outline specific future research directions, such as the need for longitudinal studies to monitor the evolution of fHbp variants over time, the development of novel vaccines targeting the most prevalent and persistent variants, and the implementation of genomic surveillance programs to detect emergent variants. Suggesting how these findings could guide public health strategies and vaccine policy would provide valuable insights for future efforts in meningococcal disease prevention.

Response: Thank you for summarizing our research and pointing out future research directions. We have added a new discussion section based on your summary suggestions as the final part of our discussion, see lines 421 to 431:

“Our research outlines specific future research directions. First, it is imperative to standardize laboratory operations across all levels, enhance monitoring, isolate bacterial strains, and conduct whole-genome sequencing. In monitoring antigen genes like fHbp, it is essential to utilize whole-genome data for improved reproducibility and traceability, with gene diversity analysis relying on full coding sequences of antigen genes rather than partial ones. Second, continuous, long-term, and cross-regional data are crucial for monitoring highly variable antigen genes like fHbp. While new variants emerge, the majority are transient; thus, a foundation for vaccine development and immunization planning can only be established through sustained monitoring. Third, the development of meningococcal B vaccines in China should prioritize the major domestic fHbp variants of serogroup B strains.”

Comments 5: A thorough proofreading could enhance the manuscript by identifying and correcting any minor grammatical errors, typographical mistakes, or inconsistencies in terminology usage. Attention to detail in these areas will further improve the overall quality and readability of the document. For instance, "Factor H Binding Protein" is sometimes referred to as "fHbp" and other times in its full form. Decide on one format and stick to it for the entire document to maintain consistency.

Response: Thank you for your suggestion. We have carefully checked the manuscript, and found that "factor H binding protein" appears only twice in the abstract and main text respectively. Throughout the manuscript, we consistently use "fHbp" as the abbreviation for "factor H binding protein", except for the vaccine name MenB-FHbp, for which we retain its original form.

Comments 6: Review the manuscript for proper use of hyphenation in compound adjectives. For example, "high genetic diversity" does not require a hyphen, but compound modifiers like "long-lasting fHbp variants" do. Ensure these rules are uniformly applied.

Response: Thank you for your suggestion. We have thoroughly examined the manuscript for hyphen usage and have confirmed that the term "high genetic diversity" does not contain a hyphen in the manuscript.

Comments 7: Check for proper use of commas in compound sentences and ensure that clauses are correctly separated. For example, "In order to comprehensively investigate the genetic diversity and epidemiological patterns of fHbp variants within isolates of Chinese N. meningitidis, we utilized the NEIS0349 locus..." The comma after "meningitidis" correctly separates the introductory phrase from the main clause.

Response: Thank you for your suggestion. We have checked the use of commas in clauses throughout the manuscript to ensure that there are no grammar errors.

Comments 8: Upon the first mention of an acronym, the term should be spelled out with the acronym in parentheses. Subsequent mentions can use the acronym alone. Verify that this rule is consistently applied, especially for terms like NEIS0349, MLST (Multilocus Sequence Typing), and others relevant to your study.

Response: Thank you for your suggestions. We have thoroughly reviewed all abbreviations with the exception of NEIS0349. NEIS0349 is the designated name for the fHbp gene within the core genome multilocus sequence typing dataset for Neisseria meningitidis in the pubMLST database (https://pubmlst.org/organisms/neisseria-spp/fhbp). Despite our efforts, we have not found a non-abbreviated name for NEIS0349.

Round 2

Reviewer 2 Report

Comments and Suggestions for Authors

Accept as it is.